# Structure and Distribution of Health Care Costs across Age Groups of Patients with Multimorbidity in Lithuania

**DOI:** 10.3390/ijerph18052767

**Published:** 2021-03-09

**Authors:** Laura Nedzinskienė, Elena Jurevičienė, Žydrūnė Visockienė, Agnė Ulytė, Roma Puronaitė, Vytautas Kasiulevičius, Edita Kazėnaitė, Greta Burneikaitė, Rokas Navickas

**Affiliations:** 1Faculty of Medicine, Vilnius University, Čiurlionio Str. 21, LT-03101 Vilnius, Lithuania; laura.nedzinskiene@mf.vu.lt (L.N.); Elena.Jureviciene@santa.lt (E.J.); Zydrune.Visockiene@santa.lt (Ž.V.); Roma.Puronaite@santa.lt (R.P.); Vytautas.Kasiulevicius@santa.lt (V.K.); Edita.Kazenaite@santa.lt (E.K.); Greta.Burneikaite@santa.lt (G.B.); 2Vilnius University Hospital, Santaros Klinikos, Santariškių Str. 2, LT-08661 Vilnius, Lithuania; 3Epidemiology, Biostatistics and Prevention Institute, University of Zurich, Hirschengraben 84, 8001 Zurich, Switzerland; agne.ulyte@uzh.ch; 4Faculty of Mathematics and Informatics, Institute of Data Science and Digital Technologies, Vilnius University, Naugarduko g. 24, LT-03225 Vilnius, Lithuania

**Keywords:** care costs, patients with multimorbidity, Lithuania

## Abstract

Background. Patients with multimorbidity account for ever-increasing healthcare resource usage and are often summarised as big spenders. Comprehensive analysis of health care resource usage in different age groups in patients with at least two non-communicable diseases is still scarce, limiting the quality of health care management decisions, which are often backed by limited, small-scale database analysis. The health care system in Lithuania is based on mandatory social health insurance and is covered by the National Health Insurance Fund. Based on a national Health Insurance database. The study aimed to explore the distribution, change, and interrelationships of health care costs across the age groups of patients with multimorbidity, suggesting different priorities at different age groups. Method. The study identified all adults with at least one chronic disease when any health care services were used over a three-year period between 2012 and 2014. Further data analysis excluded patients with single chronic conditions and further analysed patients with multimorbidity, accounting for increasing resource usage. The costs of primary, outpatient health care services; hospitalizations; reimbursed and paid out-of-pocket medications were analysed in eight age groups starting at 18 and up to 85 years and over. Results. The study identified a total of 428,430 adults in Lithuania with at least two different chronic diseases from the 32 chronic disease list. Out of the total expenditure within the group, 51.54% of the expenses were consumed for inpatient treatment, 30.90% for reimbursed medications. Across different age groups of patients with multimorbidity in Lithuania, 60% of the total cost is attributed to the age group of 65–84 years. The share in the total spending was the highest in the 75–84 years age group amounting to 29.53% of the overall expenditure, with an increase in hospitalization and a decrease in outpatient services. A decrease in health care expenses per capita in patients with multimorbidity after 85 years of age was observed. Conclusions. The highest proportion of health care expenses in patients with multimorbidity relates to hospitalization and reimbursed medications, increasing with age, but varies through different services. The study identifies the need to personalise the care of patients with multimorbidity in the primary-outpatient setting, aiming to reduce hospitalizations with proactive disease management.

## 1. Background

With all scientific advancements, increasing economic wellbeing, most people within developed countries are living longer, but at what cost? The Health Profile for England 2018 report summarises the trends of longer life, while also increasing the number of years living with disability and multimorbidity. In fact, in recent decades, the number of people with non-communicable diseases has been increasing dramatically, mostly due to the ageing population [1]. The trend is global, projecting that children under 5 (67,628 million) will be outnumbered by those over 65 (67,807 million) globally. With increasing life expectancy, the prevalence of multimorbidity (MM) is also rising. MM, defined as two or more concurrent chronic diseases, is estimated to range from 13% to 95% depending on the specific age and population group [2], and the proportion is higher in older populations [3,4]. Due to the progressive increase in the proportion of elderly people, healthcare systems are facing serious organizational and financial challenges [5,6]. Cardiovascular diseases, cancer, diabetes mellitus, and dementia are the most common chronic diseases among older adults [7], leading to impaired physical function, dependence, high care costs, and shorter survival [1]. There is well-established evidence, that healthcare costs are associated with increasing age. Some studies argue, that that proximity to death is a more important determinant of health expenditure than ageing alone. However, there is ample evidence in the literature of a positive association between MM and health care costs [8,9,10]. Costs significantly increase with each additional chronic disease due to the high need for physician, hospital and social services. older adults living with multimorbidity use significantly more prescription medications and have higher prescription drug expenditures [8].

The costs of MM were calculated in a number of patient cohorts in Germany [11], Switzerland [12], Taiwan [13], and the United States (US) [14,15], but not in nationwide studies. Analyses showed disproportionately higher direct costs for healthcare services and indirect costs of absenteeism and reduced workforce participation [10,14]. For instance, in the US, one in five patients have multiple chronic diseases; however, they account for 78% of health expenditure [15]. Therefore, patients with multimorbidity constitute a target population for which prevention could result in the biggest savings.

The health care system in Lithuania is based on mandatory social health insurance, financed by salary contributions paid by employers and employees to the Republic Health Insurance Fund. State funds cover medical services such as primary care, specialist treatment, hospitalisation, prescriptions, services during pregnancy and childbirth, and rehabilitation. Our previous studies showed a significant increase in the prevalence of MM in the Lithuanian adult population reaching 62.30% at the age of 85 years and older with a steep rise at the age of 45–55 years [16]. We also identified age breaking points and showed that MM starts to increase at 28 years and grows more rapidly in younger patients than in the elderly, and these relatively young patients with MM use more health care resources for expensive medications and hospitalizations compared with other same-age patients [17]. However, a comprehensive assessment of health care costs in different age groups, its association to MM complexity in patients with multimorbidity in Lithuania is still lacking.

Therefore, our current study aimed to explore the distribution, change and interrelationships of health care costs across the age groups of patients with multimorbidity in Lithuania.

## 2. Methods

### 2.1. Study Population

The data from the Lithuanian National Health Insurance Fund (NHIF) database, covering the period from 1 January 2012 to 30 June 2014 of adult patients (18 years and older) with two or more chronic diseases were analysed. The identified period was chosen due to the reliability of the data, which was achieved by updating national rules and regulations for medical coding starting in 2012. The NHIF database encompasses about 98% of inpatient cases and 90% of outpatient visits (up to 100% of primary healthcare visits) in Lithuania, covering the entire territory of the country and having about 7000 users (healthcare institutions) working with the system. The database contains demographic data and entries on the primary and secondary healthcare services provided, emergency and hospital admissions, and prescriptions of reimbursed medications for chronic diseases. For missing data, average values were used. Each visit or service is associated with a corresponding standardised cost that is established by the state. In this sense, the cost of a visit or hospitalization was calculated as the total of the standard costs of all component services provided.

Data exporter software was used to extract the following data: demographic information (sex, age) of patients who were diagnosed with at least one of 32 chronic diseases (Table 1); the use of primary (overall and home visits), outpatient (specialist consultation) healthcare services; hospitalizations; prescribed and at least partially reimbursed medications during the analysed period. Patient’s age was defined as the age in the year of data extraction and was classified into 8 categories: from 18 to 24 years, from 25 to 34 years, from 35 to 44 years, from 45 to 54 years, from 55 to 64 years, from 65 to 74 years, from 75 to 84 years, and aged 85 years and over. The number of chronic diseases, mean cost of medication, hospitalization, primary and outpatient services, the share of health care costs and the trends of annual costs of a patient with multimorbidity were assessed in different age groups. The anonymised information was uploaded on the secured server which was developed for this study. The analysis of the anonymised data was allowed according to the Lithuanian data protection regulation without the explicit consents of the patients.

### 2.2. Statistical Analysis

Patients aged 18 years and over with at least 2 chronic diseases from the list in Table 1 during the 2.5 years period were selected for the analysis. Patient characteristics are presented as means ± standard deviation (SD) for continuous variables and as frequencies and percentages for categorical variables. The 95% confidence intervals (CI) were computed for health care service costs. Generalised linear models (GLM) with gamma distribution and log link function were used for evaluation of the relationship between chronic diseases and cost. The segmented linear regression was used to identify breaking points (R package “segmented” [18]). For statistical analysis, R statistical computing environment (version 3.4.0; R Core Team 2017) [19] and STATISTICA (StatSoft, version 10) were used. All reported *p* values were two-tailed and the level of significance was set at 0.05.

## 3. Results

### 3.1. Study Sample and Health Care Service Cost

The sample of patients with at least one chronic disease included 452,769 patients—273,016 (60.3%) women and 179,753 (39.7%) men. Of these, 428,430 had two and more chronic diseases were classified as patients with multimorbidity and thus included in the further analysis. The average age of all included patients was 67.83 (SD = 13.34) years, females—69.71 (SD = 13.00) years, males—64.91 (SD = 13.33) years. Patients with multimorbidity had on average 4.79 ± 2.05 chronic diseases, with the highest average number of chronic conditions of 5.25 ± 2.08 in the age group of 75–84 years. Notably, even in the youngest age group (18–24 years), the mean number of chronic diseases reached 2.52 ± 0.84, gradually increasing with age and almost doubling at the age of 65–74 years.

The total cost of 302,843,036.21 EUR of all patients with multimorbidity per year was composed of primary care cost—14,194,613.70 EUR (4.69%), outpatient cost—13,017,573.87 EUR (4.30%), hospitalization cost—156,078,078.78 EUR (51.54%), and medication cost 119,552,769.86 EUR (39.48%). Of them, 93,590,567.02 EUR (78.28%) were spent for reimbursed, 26,834,777.81 EUR (22.46%)—paid out-of-pocket medication). More than half of all expenses (51.54%) were allocated for hospitalization costs and another big part (30.90%) for reimbursed medications.

Total health care services cost for a patient with multimorbidity was on average 707.15 EUR (95% CI 703.37–710.93 EUR) annually. Although there was a higher proportion of women in the cohort, the annual cost was significantly higher for male patients—756.52 EUR (95% CI 749.95–763.09 EUR) vs. 631.28 EUR (95% CI 627.13–635.42 EUR), respectively (*p* < 0.001), and for patients coming from urban compared to rural areas-688.79 EUR (95% CI 683.90–693.68 EUR) vs. 674.59 EUR (95% CI 668.0–681.18 EUR) respectively (*p* < 0.001).

### 3.2. Health Care Cost and Age

The assessment of the annual cost of a patient with multimorbidity showed different spending for the specific health care services in different age groups. The highest average total amount of 797.59 EUR and the highest hospitalization cost of 443.00 EUR was spent in 75–84 years, the highest cost of medication was 304.64 EUR in 65–74 years, the highest cost for primary care was 36.12 EUR in 65–85+ years, and the highest cost for outpatient care was 34.76 EUR in 45–54 years age group (Table 2).

Further analysis of total average cost and the share of all spending for primary and outpatient visits, hospitalization and prescribed medications calculated as a function of age, showed variably increasing trend throughout age groups and different health care services (Figure 1). The average total cost increased linearly and steadily from age 18 to 75 years, was somewhat stable for older age and then decreased in the eldest age group of 85+ years. Services, constituting the biggest share of the average cost, were different depending on the age group.

Hospitalization costs amounted to the highest cost for age from 18 to 22 years and 40 years on (with the minimum cost reached in the age of around 25 years), then increased steadily until the age of around 79 years and stabilised again. The average cost of medications increased in a similar steady fashion until the age of around 72 years and stabilised afterwards. The cost of medication paid out-of-pocket increased more markedly than the reimbursed cost between the age of 28 and 53 years. Importantly, although the price of medications and hospitalization was comparable for age groups up to around 50 years, the rise of hospitalization cost was steeper, resulting in about 1.6 times higher average cost for 85+ years age group. The average outpatient cost was fairly stable for age groups from 18 to 75 years and decreased from 75 years on.

In addition to health care cost trends, we estimated the age breaking points to identify the age when the costs for different health care services start to increase, decrease or become stable. Only one age breaking point indicated a decrease in health care expenses for hospitalization at 22.09 years. All other age breaking points indicated either stable or increased expenses and the start of changes was noted in quite young patients-increased outpatient costs from age 26.53, out-of-pocket expenses from 27.82, hospitalization costs from age 38.64. At each age breaking point, each new illness increased costs 1.26 times (Figure 1).

The distribution of age group shares in the total spending for patients’ with multimorbidity health care was similar for all services analysed. For all service categories, the relative share spent on age groups from 18 to 44 years was negligible, accounting for 3.10% of total spending and ranging from 0.13% for primary, 0.23% for outpatient, 1.33% for hospitalization to 1.40% for medication cost. The share in the total spending gradually increased for older age groups, and peaked in the age group of 65–74 years, accounting for 29.19% (1.39% for primary, 1.30% for outpatient 14.77% for hospitalization and 11.73% for medication costs), and 75–84 years, accounting for 29.53% (1.34% for primary, 0.99% for outpatient, 16.35% for hospitalization and 10.85% for medication costs) (Figure 2).

As the number of patients in each age group differed (ranging from 2000 in 18–24 years to 116,423 in 65–74 years), we have estimated the average amount spent in EUR per patient for one year for different health care services and plotted against the proportion of expenses in age groups (Figure 3).

Although there was a gradual steep increase in expenses per patient with the increasing age and proportion of expenses almost in all age groups, this tendency was seen up to the age of 76–84 years. There was a decrease in expenses per patient after the age of 85+. The only service with expenses per patient stable from 65 years on was primary care. At the same time, there was a decrease in the expenses for outpatient service and an increase in hospitalization, signalling that elderly patients tend to approach hospital emergency departments rather than outpatient clinics.

## 4. Discussion

About 50 million people live with multiple chronic diseases in the European Union (EU) and they use 70–80% of total health care resources [20]. We aimed to assess the structure of health care costs in different age groups of patients with multimorbidity in Lithuania (a member of the EU) and revealed that about 60% of the total cost is attributed to the age group of 65–84 years. We have also estimated a steep increase in hospitalization costs starting at the early age of 45 and peaking at 75–84 years, while at the same time–decreased costs for outpatient care in the age group of 75 years and older. The number of chronic diseases was increasing with age and was the highest in 75–84 years reaching the mean of 5.25 (SD = 2.08). These results can partially explain high health care cost and are in line with previous research. In the United States in 2005, the increases in average Medicare costs per beneficiary associated with 1, 2 and ≥3 chronic diseases were $7172, $14,931 and $32,498, respectively [21]. In Ireland, the adjusted per-capita total healthcare costs increased from 760.20 EUR for 0 to 4096.86 EUR for >4 chronic diseases [22].

To understand the factors leading to this increase we analysed the structure of expenses and estimated that the highest proportion of all expenses in Lithuania is used for hospitalization and medication reimbursement in all age groups. The total cost for patients with multimorbidity in Lithuania was 302,843,036.21 EUR per year. More than half of all the expenses were allocated for hospitalization costs and more than 30% for reimbursed medications. Hospitalization costs in Lithuania were different in age groups and amounted to the highest proportion of cost at the age of 18 to 22 years and from 40 years on, then increased steadily until the age of around 79 years, and stabilised again. The impact of MM on hospitalization costs was analysed in numerous studies. In Germany, patients with multimorbidity aged 72 years and above used a substantial range of health care resources of which 25% were due to inpatient care, 20%—to outpatient physician services and 20%—to medications [11]. A study in Switzerland, with the mean age of the study sample at 74.9 years, showed that 50.80% of the total costs accounted for outpatient services, 24.10% for inpatient services, and 25.10% for medication costs [12].

A cross-sectional survey among Singapore residents aged 60 years and above showed that the costs of hospitalization, contacts with hospital doctors, and medication were the biggest drivers of health care costs and were three to seven times higher for the patients with multimorbidity compared to those with none or one chronic disease respectively [23]. A retrospective cohort study in the province of Ontario showed that hospitalization among individuals <65 years was the primary cost driver and responsible for 47% of total healthcare costs, followed by the physician (32%), drug (10%) and continuing care costs (6%). For older adults, hospital costs remained the largest cost component (41%), followed by continuing care (23%), drug (19%) and physician costs (15%) [24]. Although our data are in line with previous research (hospitalization costs account for a substantial amount of total expenses), this proportion is very high in our study. This could be due to several reasons, one of which—a clear tendency of reduced use of outpatient and stable use of primary care service from the age of 65 years, when one would expect these services to be used more frequently with the ageing and an increasing number of chronic diseases. Current health policy in Lithuania promotes the active involvement of family doctors in the care of patients with multimorbidity and strongly suggests to consider individually whether the patient needs to be referred to the specialist to avoid unnecessary consultations. However, it is clear from our data, that reduced referral to specialists, reflected by reduced expenses for this service, is not an optimal solution. Another important point to consider is that the budget spent on primary and outpatient care was about 11 times lower, compared to the expenses on hospital care. This strongly suggests that current outpatient care is insufficient and inappropriately funded for patients with multimorbidity. Improvement in outpatient care management, implementation of an integrated health care model would hopefully allow to optimise the use of health care resources, reduce hospitalization costs and improve patients’ health.

We estimated that the expenses for medication cost account for 39.60%, with the cost for reimbursed medications 30.90% and cost for medications paid out-of-pocket 8.70% of the total budget. The mean annual cost for prescription medications has been analysed previously. Fahlman et al. showed a significant increase in spending with each additional chronic disease, Moxey et al. also observed the same tendency, comparing subjects with zero, one or two, and three and more chronic diseases, Sambamoorthi et al. associated mean increase of costs for medications by more than 2.5 times in subjects with two or more compared with subjects without chronic diseases [25,26,27].

Detailed analysis across age groups is lacking in the literature, although particularly among elderly patients, individuals face not only a financial burden of the cost for reimbursed medications, but also the substantial increase in the cost paid out-of-pocket. Patients of 65 years and older are at the risk of a lower-income and the possibility of non-compliance with prescribed treatment. Polypharmacy due to the increasing number of chronic diseases, possible adverse drug events and fragility—all together explain why effective care in patients with multimorbidity is not achieved irrespectively of high spending.

An interesting tendency observed in our study was the decrease in health care expenses per capita in patients with multimorbidity after 85 years of age. This was observed for all services except primary care, where expenses remained stable, compared to the age group of 65–84 years. To explore and explain this tendency we would need a more detailed analysis of specific chronic diseases, the severity of diseases, demographic characteristics and social care in this particular age group. Although similar results were obtained in the Basque Country study where health care expenditures were concentrated in patients with multimorbidity, and the biggest part in the age group of 70–84 years, detailed information on the elderly population is lacking [28].

We found that annual expenses for health care were almost 20% higher in men than in women. This difference might be related to the prevalence of different disease clusters, the perceived importance of health in men and women or socio-demographic factors and inequality. According to Statistics Lithuania, the average life expectancy is 69.5 years for men and 80.0 years for women [29]. Therefore, we cannot explain lower expenses for women by the age factor. This finding differs from other studies. In the Ontario study, expenses for women of 65 years and older were higher and were partially explained by longer life expectancy and greater risk of MM in older women than men [24]. In other studies, increased costs of medications and out-of-pocket payment were estimated for women [25,30]. Thus, more research is needed to assess men and women differences in multimorbid population in Lithuania.

There are numerous other factors influencing health care expenses and we face a challenge in measuring costs of MM [31]. Although we have estimated that each additional chronic disease increases cost by 1.26 times, we are aiming for further research to use a multifactorial approach for a more comprehensive situation analysis.

The use of the NHIF administrative database has a few limitations. Even though the database is very detailed, this database is created for the reimbursement of health care institutions for the medical services provided by the National Health Insurance Fund as well as for statistical needs. The administrative database includes inpatient, outpatient events, as well as prescription medication and other direct healthcare-related costs(reimbursed). The information is based on documentation from the health care providers. Although a uniform system for diagnoses reporting exists, doctors may still under-report or over-report chronic diseases or any events, relating to the disease for various reasons, including funding related ones. To minimise this bias we have used a data collection period of 2.5 years presuming that during this time chronic diseases will be recorded at least once and thus, included in our analysis. A national database, including all healthcare providers, was included, limiting the factor of unfair coding in some institutions or teams. Additionally, comprehensive database information relating to single patient diagnoses, treatment, primary and outpatient visits and hospitalizations, as well as their trends over time, increase the reliability of our data. The list of chronic diseases used to select patients (Table 1) was based on a literature review and was described previously [32], however it is debatable and could possibly be expanded. Different authors may propose a different list of chronic disease, some excluding mental illness, accounting for worse outcomes and increased costs. Due to a lack of an established multimorbidity gold standard measure, defining the list of diseases, it is impossible to clearly define which chronic diseases should be excluded, as clearly some of the minor ones would have a different weight on the outcomes, costs, and management of the patients. However, we have analysed the prevalence of selected chronic diseases in the Lithuanian population in our previous work and confirmed that the chronic conditions included in the current study’s definition are highly prevalent and thus meeting the criteria of chronic diseases having the highest impact on the use of health care resources [16], limiting ourselves to a previously defined multimorbidity disease list.

The duration of the database was chosen to cover more than 2 years, reducing the chance the patient never visited the healthcare provider. However, the period might not be long enough to cover the whole picture of some diseases, more prone to fast deterioration. The limited duration and scope of the database do not allow for further analysis of the association with the end of life care. The size of the database means the duration had to be limited. It also affected the time frame of the data, suggesting some changes in the care might have happened during the last few years, drastically affecting the resource usage, even though, very unlikely. Last, but not least, the analysis is limited to a single country. Additional database analysis of other countries could potentially validate the findings, suggesting the need for further analysis, supporting the differentiation in the reimbursement of services, for the investigated patients with multimorbidity group.

## 5. Conclusions

Healthcare expenses for patients with multimorbidity in Lithuania gradually increase up to the age of 65 years, remain fairly stable till 84 years and then decrease. The highest proportion of all expenses was for hospitalization costs and medication reimbursement in all age groups. There was a clear tendency of reduced use of outpatient service with the increased use of hospitalizations from the age of 65 years, which also reflected in the increase in total health care expenses. The study identifies the need to optimise the care of patients with multimorbidity by implementing a complex model in the primary-outpatient setting. Currently, such an integrated model addressing specific needs in the Lithuanian health care system is under development and implementation.

## Figures and Tables

**Figure 1 ijerph-18-02767-f001:**
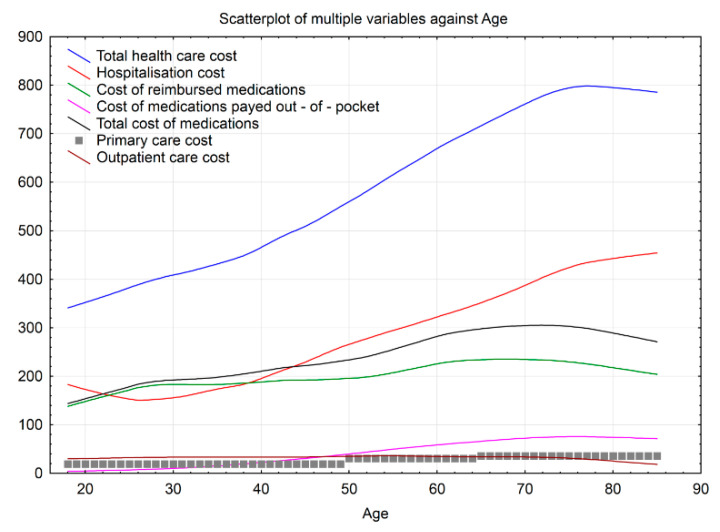
Trends of the mean annual cost of a patient with multimorbidity in different age groups.

**Figure 2 ijerph-18-02767-f002:**
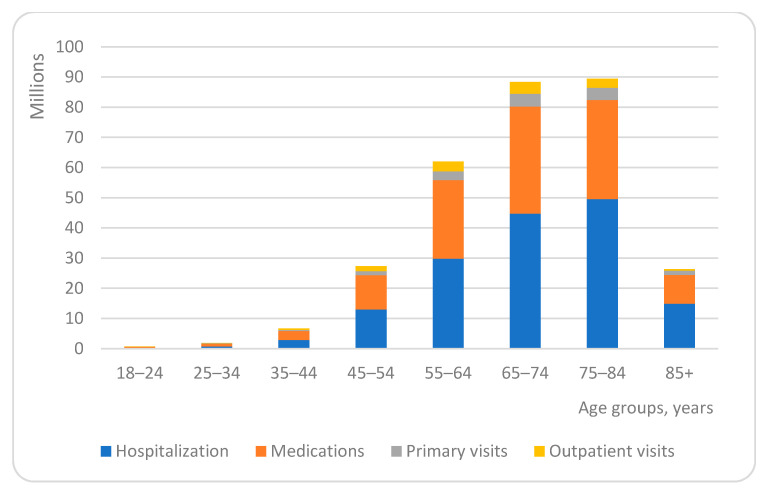
Share of health care costs in different age groups.

**Figure 3 ijerph-18-02767-f003:**
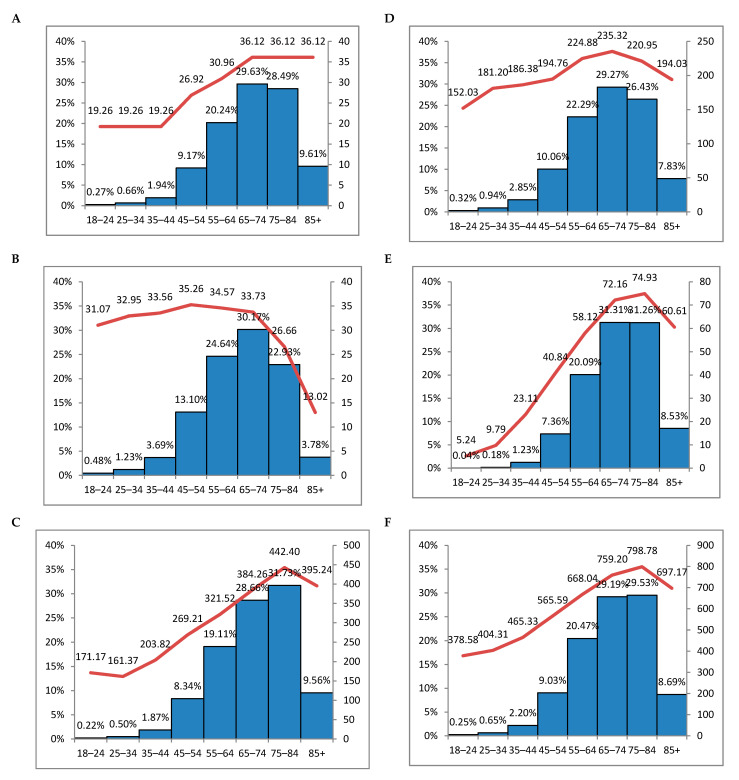
The average amount spent in EUR per person for one year plotted against the proportion of expenses in age groups: (**A**)—primary care cost; (**B**)—outpatient cost; (**C**)—hospitalization cost; (**D**)—the cost of reimbursed medications; (**E**)—the cost of medications paid out of pocket; (**F**)—total cost (**A** + **B** + **C** + **D** + **E**). Bars—proportions of expenses in age groups; line—the average amount in EUR per person per year.

**Table 1 ijerph-18-02767-t001:** The list of the selected chronic diseases associated with ICD-10-AM diagnostic codes.

Chronic Diseases with ICD-10-AM Diagnostic Codes	
1	Cancer C00–C96
2	Anaemia D50
3	Hypothyroidism E02; E03; E89.0
4	Diabetes E10.0–E10.9; E11.0–E11.9
5	Obesity E66
6	Dyslipidaemia E78
7	Dementia F00.0–F00.9; G30.0–G30.9; F01.0–F01.9; F02.0–F02.8; F03
8	Mental disorders F20.0–F20.9; F30.0–F39; F40.00–F40.9; F41.0–F41.9; F42.0–F42.9; F43.0–F43.9
9	Parkinson disease G20
10	Multiple sclerosis G35
11	Epilepsy G40.00–G40.91
12	Sleep apnoea G47.3
13	Back Pain G54.1; G54.4; G55.1; M51
14	Glaucoma H40–H42
15	Blindness H53–H54
16	Hearing loss H90.0–H90.8; H91.0–H91.9
17	Hypertension I10–I15
18	Ischemic heart disease I20–I25
19	Arrhythmias I44–I49
20	Heart failure I50.0–I50.9
21	Intracranial bleeding I61–I62
22	Stroke I63-I64; I69
23	Chronic obstructive pulmonary disease J44.0–J44.9; J96
24	Asthma J45.0–J45.9
25	Inflammatory bowel disease K50; K51
26	Psoriasis L40.0–L40.9
27	Rheumatoid arthritis M05–M06
28	Gout M10.0–M10.99
29	Osteoarthritis M15–M19
30	Systemic lupus erythematosus M32
31	Osteoporosis M80–M82
32	Renal failure N18–N19

**Table 2 ijerph-18-02767-t002:** The annual cost of a patient with multimorbidity in EUR in different age groups.

Age, Years	N	Mean of Total Amount	Mean Cost of Medication	Mean Cost of Hospitalization	Mean Cost of Primary Care	Mean Cost of Outpatient Care
18–24	2000	378.70	154.64	173.83	19.26	30.97
25–34	4861	401.72	190.17	159.29	19.26	33.00
35–44	14,305	455.38	206.85	195.45	19.26	33.82
45–54	48,357	554.47	231.46	263.14	25.11	34.76
55–64	92,777	665.05	279.58	319.79	30.96	34.71
65–74	116,423	757.05	304.64	382.54	36.12	33.75
75–84	111,953	797.59	292.40	443.00	36.12	26.07
85+	37,754	697.17	252.78	395.24	36.12	13.02

## Data Availability

The data that support the findings of this study are available from Lithuania National Health Insurance Fund but restrictions apply to the availability of these data, which were used under license for the current study, and so are not publicly available. Data are however available from the authors upon reasonable request and with permission of Lithuania National Health Insurance Fund.

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
