# Peer review of "Structure and Distribution of Health Care Costs across Age Groups of Patients with Multimorbidity in Lithuania"

_ijerph, 2021, doi:10.3390/ijerph18052767_

Round 1

Reviewer 1 Report

Here, Laura et al. conducted a statistic study that systematically analyzed the health care resources distribution among different age groups of multimorbid patients in Lithuania. The authors divided the patients based on their age and listed their annual cost, analyzed the trends of mean annual cost, summarized the share of health care cost et al. This work could provide insightful information for optimizing the health care system in Lithuania. The data is well analyzed and organized. The overall writing of the manuscript is good but some sentences are too long to read. Some minor word editing is necessary. Otherwise, I think the manuscript is ready to accept.

Author Response

Dear Reviewer,

Thank you very much for your revision and kind words. Some minor word editing was done. In order to make the manuscript more reader friendly, we have made some adjustments to the structure of the sentences. 

Reviewer 2 Report

Summary:
In the paper entitled with "Structure and distribution of health care costs across age groups of multimorbid patients in Lithuania", the authors
present the results of a study in Lithuania with the goal to investigate costs of multimorbid patients from different perspectives and on a nationwide
scope. Relevant background information and related works are discussed. The authors describe the data source of the analysis as well as the methods
used to analyse the data. Then, the results of the investigation are presented and discussed, including a detailed discussion of limitations. The
authors conclude with detailed results on what should be done in Lithuania and what findings are interesting in general.

Points in favor of the paper.
- The paper fits to the scope of the journal
- The paper draws a clear contribution
- The paper deals with a relevant topic
- The paper shows experimental results
- The results are based on a very large data source
- The paper results are sound
- The paper discusses limitations explicitly and comprehensively

Points against the paper:
Although I like the paper, some aspects should be improved:
- The abstract is too long
- A thorough proofreading is needed, there are many language issues in the paper, e.g.,
(1) First sentence of the abstract is not a correct sentence
(2) Abstract: "... at different age group" -> "... at different age groups."
(3) Page 2, Line 47: "... the trends, of longer life ..." -> "... the trends of longer life ..."
...
...
- In general, commas should be checked
- Page 3, Line 115: JA-CHRODIS server is mentioned and never explained
- Page 4, Line 129: p-values are mentioned, but no p-value is reported?
- It would be nice to have legends in Figures 3

Author Response

Dear Reviewer,

thank you for your time and efforts you have invested during this revision. Thank you for revising the article and for providing useful aspects that could be improved. We highly appreciate your reccomendations and have taken into account all your suggestions. We have corrected the final version accordingly. We have revised the abstract and checked commas. Some minor word editing was done. In order to make the manuscript more reader friendly, we have made some adjustments to the structure of the sentences.

In regards of the comment on "Page 3, Line 115: JA-CHRODIS server is mentioned and never explained", we have corrected the sentence and clarified that the data was uploaded “…on the secured server which was developed for this study”.

We have also clarified that all reported p values were two-tailed and the level of significance was set at 0.05.

In addtion to that, as requested, we have included the description of legends in Figures 3.

Reviewer 3 Report

An interesting complement to the author group's previous two publications. Might benefit from more discussion to tie the findings for Lithuania over the time period of these studies and the evolution of data and data sharing (health information exchange): 2015, 2018.

Author Response

Dear Reviewer,

thank you for your time and efforts you have invested during this revision and thank you for your kind comments. Authors are working on the next articles, which will include updated data and further interesting findings. 

Reviewer 4 Report

Thank you for the opportunity to review this manuscript entitled "Structure and distribution of health care costs across age groups of multimorbid patients in Lithuania" that aimed to explore the distribution, change and interrelationships of health care costs across the age groups of patients with multimorbidity.  I have included suggested revisions below and I would be happy to review a resubmission if necessary. 

OVERALL
-Please ensure the use of the term "patients with multimorbidity" instead of "multimorbid patients" throughout, including in the manuscript title 
-Please ensure more consistent use of the abbreviation "MM" throughout
-Please ensure consistent use of either the term "diseases" or "conditions" (not both) throughout 
-Please ensure consistent use of either one or two decimal points throughout
-Please clarify why a stratified analyses between females and males, as well as between types of combinations of chronic conditions, was not conducted within the current study 
-As well, please clarify why the definition of two or more chronic conditions was selected (as compared to three or more chronic conditions) and why stratified analyses was not conducted based on the number of chronic conditions a patient was living with, as opposed to primarily stratifying by age group 

ABSTRACT 
-Line 17 - Please remove comma after "Even though"

MAIN TEXT
-Line 19 - Please change "communicable disease" to "communicable diseases" 
-Line 24 - Please change "age group" to "age groups" 
-Line 24/25 - Please change "screened all adult population with at least single event for NCD when" to "identified all adults with at least one chronic conditions when"
-Line 26 - Please change "single chronic condition owners" to "patients with single chronic conditions" 
-Line 30/31 - Please change "study screened all adult population in Lithuania, incuding 428.430 adults into more detailed analysis" to "study identified a total of 428,430 adults in Lithuania"
-Line 37 - Please change "recorded" to "observed" 
-Line 41 - Please change "hospitalisations" to "hospitalizations"

BACKGROUND
-Line 45 - I would suggest rephrasing this first sentence to better reflect that most people within most developed countries are living longer
-Line 47 - Please remove the comma after "the trends"
-Line 47 - Please clarify what is meant by "the life expectancy in the poor health", such as a life expectancy that involves the increased number of years living with disability and morbidity? 
-Line 50 - Please include a reference of the number of children under 5 being outnumbered by those over 65 years
-Line 57 - Please change "in elderly" to "among older adults"
-Line 59 - Please remove the capital from "healthcare"
-Line 60/61 - Please include a reference to the studies arguing that the proximity of death is a determinant of health expenditure 
-Line 64 - As noted, please replace "elderly multimorbid patients" to "older adults living with multimorbidity" 
-Line 68 - Please change "huge" to another descriptor like "higher" 

METHODS 
-Line 94/95 - Please change "provided" to "identified" 
-Line 96 - Please change "from 2012" to "in 2012"
-Line 99 - Please clarify in brackets what type of "users" work with the system
-Line 103/104 - Please clarify what a "hospitalization period" means or remove the term "period" 
-Line 106 - Please change "of people," to "of patients who were"
-Line 109 - Please change "defined" to "calculated"
-Line 110/111 - Please add "years" after each category 
-Please describe how missing data were addressed in this analysis, specifically for age and sex data

RESULTS 
-Line 133/134 - Please replace the decimal points with commas 
-Line 133 - Please change "subjects" to "patients" 
-Line 136 - Please provide a more specific average age (with standard deviaton) for both females and males than "over 67 years" 
-Line 137/138 - Please change the "highest number" to the "highest average number of chronic codnitions"
-Line 142 - Line 145 - Please include a proportion in brackets for each category of costs
-Line 159 - Please clarify if "65-85+ years" is a correct category 
-Line 167 - Please change "75" to "75 years" 
-Line 211 - Please clarify what "almost in all categories" means 

DISCUSSION
-Line 220 - Please clarify if Lithuania is a member of the European Union
-Line 225 - Please include the standard deviation for the mean number of chronic conditions
-Line 245 - Please start this new paragraph with "A"
-Line 246 - Please clarify what the term "restructured" means 
-Line 249 - Please clarify if this study was for all patients or specifically for patients with multimorbidity 
-Line 273 - Line 275 - Please remove the italics 
-Line 279 - Please change "in elderly we face" to "among elderly patients, individuals face"
-Line 283 - Line 285 - Please clarify what is meant by "adequate cost-effective care not achieved irrespectively of high spending" 
-Line 304 - Please clarify why a stratified analyses between females and males was not conducted within the current study 
-Line 320 - Please change "National database" to "A national database"
-Line 326/327 - Please change "would be proposing different chronic disease list" to "may propose a different list of chronic diseases" 
-Line 328 - Please change "to lack of multimorbidity definition" to "to a lack of an established multimorbidity gold standard measure" 
-Line 330 - Please change "excluding" to "excluded" 
-Line 333 - Please change "they" to "the chronic conditions included in the current study's definition" 
-Line 339 - Please change "does not allow to further analysis" to "does not allow for further analysis of"
-Line 345 - Please change "suggesting of further" to "suggesting the need for further" 
-Line 349 - Please change "age of 65" to "age of 65 years"

Author Response

Dear Reviewer,

thank you for your time and efforts you have invested during this revision. Thank you for revising the article and for providing useful and detailed notes how the manuscirpt could be improved. We highly appreciate your reccomendations and have taken into account all your suggestions. 

Please see the attachment with our point-by point responses. 

Thank you again!
